# STABILIZED TEST-TIME ADAPTATION OF NEURAL SURROGATES IN SIMULATION

## ABSTRACT

Machine learning is increasingly used in engineering to accelerate costly simulations and enable end-to-end design optimization, mapping inputs (e.g., initial conditions, parameters or meshes) to simulation results or design candidates. However, large models are often pretrained on datasets generated with assumptions (e.g., geometries or configurations) that may not hold at test time, resulting in significant performance degradation. Test-Time Adaptation (TTA) mitigates such distribution shifts by leveraging inputs, online and at test-time. It avoids the need for costly re-training and doesn't require access to ground truth labels. In this work we propose Stable Adaptation at Test-Time for Simulation (SATTS), a novel method to improve performance of neural surrogates at deployment. It excels in high-dimensional settings through stable feature alignment and self-calibration, by leveraging latent covariance structures. To the best of our knowledge, this is the first study of TTA in the context of simulation surrogates and generative design optimization.

## 1 INTRODUCTION

Neural surrogates have become powerful tools for accelerating Partial Differential Equation (PDE) simulations across engineering and science. They perform well when test conditions match the training data, but performance often drops on unseen configurations (geometry, material types, structural dimensions, desired and physical parameters), i.e., when the data distribution shifts. This challenge gets more pronounced in industrial simulation and design optimization, where configurations can vary widely across iterations and frequently extend beyond the ranges known a priori, at data generation and training time. In many cases, only large pre-trained surrogate models are available, making full retraining costly or impractical. Moreover, access to original training data may be limited due to portability or proprietary constraints, highlighting the need for model- and task-agnostic approaches that enable zero-shot adaptation and automated model selection during design optimization.

Distribution shifts (Quinonero-Candela et al., 2008) are a well researched topic and several directions have been explored, including domain generalization (Blanchard et al., 2021), meta-learning (Hochreiter et al., 2001; Hospedales et al., 2021), and active learning (Settles, 2009). In contrast, Test-Time Adaptation (TTA) adapts models during inference without source data and minimal training effort, making it an ideal approach for engineering tasks where rapid adaptation is essential (Liang et al., 2020; Sun et al., 2020b; Wang et al., 2021a). However, online TTA can face instabilities, as model updates are performed only once per batch, which makes the algorithm highly dependent on the parameters selected at test time. TTA has proven effective in many domains, including medical imaging, object detection, and segmentation. While most works treat classification tasks (Zhou & Levine, 2021; Niu et al., 2022; Zhang et al., 2022; Zhao et al., 2023; Kojima et al., 2022; Eastwood et al., 2021; Jung et al., 2023), comparably little research can be found for regression (Liang et al., 2024). One outstanding method is Significant-Subspace Alignment (SSA) (Adachi et al., 2025), capable of handling both classification and regression tasks. It is however restricted to one-dimensional regression outputs and depends on manual selection of feature parameters, potentially causing instability for high-dimensional data.

In this work, we introduce the first TTA framework for simulation neural surrogates and evaluate it on diverse engineering datasets subject to distribution shifts, including simulation (regression) and design optimization (generation) tasks. Our approach follows the principle of learning domain-invariant

representations (Ben-David et al., 2006; Ganin et al., 2015; Johansson et al., 2019). It utilizes latent covariance structures for distribution alignment and stabilization of automated parameter selection. More precisely, we perform TTA by aligning latent feature distributions, a common approach in source-free model adaptation (Bonneel et al., 2015; Ishii & Sugiyama, 2021; Eastwood et al., 2021; Jung et al., 2023). During test-time adaptation, target features are extracted and aligned with source feature statistics obtained during prior model training. To enhance this process, we introduce a novel feature weighting and selection mechanism, resulting in improved performance compared to existing approaches. We call this initial component ATTS, Adaptation at Test-Time for Simulation.

Secondly, we draw on insights from Unsupervised Domain Adaptation (UDA) and related communities, where *model selection* has been recognized as a key factor for achieving reliable performance and stability (Musgrave et al., 2021; Miller et al., 2021; Baek et al., 2023). In practice, although UDA and TTA enable adaptation without target data, their effectiveness often lacks robustness, particularly in engineering tasks (Setinek et al., 2025), which underscores the importance of model selection. We therefore extend ATTS by self-calibrating the level of adaptation online and at test-time using statistics from the source dataset, selected to maximize expression of latent covariant structures. This results in Stable ATTS (SATTS), as further illustrated by Fig. 1. To our knowledge, this is the first study of TTA in the field of neural simulation surrogates and combining it with model selection.

We evaluate SATTS on distribution shifts from the SIMSHIFT benchmark dataset (Setinek et al., 2025) for neural simulation surrogates and the EngiBench dataset (Felten et al., 2025) for end-to-end design generation, covering diverse configurations from realistic industrial simulation and design scenarios. Our results show that SATTS offers near *free test-time improvements* while demonstrating consistent stability across all tasks, with no decline at worst, compared to the highly unpredictable behavior of other TTA methods. Our contributions are summarized as follows:

- SATTS is the first TTA method for simulation neural surrogates. It often outperforms existing regression methods SSA, eliminates the need for feature hyperparameters and stabilizes adaptation.
- We evaluate TTA on distribution shifts in industrial settings, trying to cover diverse configurations from realistic engineering design and optimization scenarios using the SIMSHIFT (Setinek et al., 2025) and EngiBench (Felten et al., 2025) datasets.
- Since TTA for physical neural surrogates remains largely unexplored and it promises near free improvements at test-time, we identify promising opportunities for innovation at the intersection of physics and adaptive machine learning.

## 2 RELATED WORK

**Neural surrogates** have emerged as a widely used approach to accelerate traditional numerical simulation methods by providing fast approximations of the solutions. In general, surrogate models are trained on the solutions from numerical solvers, paired with the corresponding initial conditions and configurations under which they were generated, e.g., Setinek et al. (2025); Bonnet et al. (2022); Toshev et al. (2023; 2024). A particularly prominent line of work within neural surrogate modeling for PDEs is operator learning (Kovachki et al., 2021; Li et al., 2020; Lu et al., 2021; Alkin et al., 2024; Wu et al., 2024b). Such models aim to directly approximatethe solution operator that maps initial functions (conditions and input terms) to output functions.

**Test-Time Adaptation (TTA)** refers to the emerging machine learning technique of adapting a pre-trained model to unlabeled target data, directly at inference time and prior to generating predictions. For this reason, TTA has recently attracted increasing attention as it offers a (nearly) free performance gain (Liang et al., 2024). While the majority of existing TTA methods have been developed for low-dimensional classification tasks (Liang et al., 2021; Yang et al., 2021), employing methodologies such as entropy minimization (Wang et al., 2021a; Zhou & Levine, 2021; Niu et al., 2022; Zhang et al., 2022; Zhao et al., 2023) and feature alignment (Ishii & Sugiyama, 2021; Kojima et al., 2022; Eastwood et al., 2021; Adachi et al., 2023; Jung et al., 2023), recent works have begun to extend these ideas to image segmentation (Valanarasu et al., 2023; He et al., 2021; Karani et al., 2021). Research in regression problems is very sparse, and standard TTA methods cannot be trivially applied. One potential reason is the use of Mean Squared Error in regression problems, which often leads to a focus on a narrow set of predictive features, reducing diversity (Zhang et al., 2023). Significant-Subspace

Alignment (SSA) (Adachi et al., 2025) addresses these limitations by manually selecting and aligning the important feature dimensions, and shows positive performance in the one-dimensional cases. Finally, TTA should not be confused with Test-Time Training (TTT), often used in time series literature (Sun et al., 2020a; Wang et al., 2021b; Sun et al., 2025; 2020c). While both solve the same problem, TTT typically refers to methods that employ time-series specific techniques, for example, updating hidden states during sequential inference.

**Domain generalization, meta-learning, and active learning** represent alternative strategies that can be used to improve model robustness and generalization under distribution shifts. Domain generalization (Muandet et al., 2013; Li et al., 2017) and Unsupervised Domain Adaptation (UDA) (Sun & Saenko, 2016; Gretton et al., 2006; Zellinger et al., 2019; Ganin et al., 2015) can be effective in some scenarios, however their reliance on specific training, model selection and diverse training distributions limits their applicability. Meta-learning methods (Finn et al., 2017) and active learning (Lewis & Gale, 1994; Musekamp et al., 2025) are similarly motivated, but generally assume access to ground-truth information in the shifted domain. In our setting, all these approaches face a significant practical limitation: none of them can quickly adapt a pre-trained model leveraging unlabeled data at test-time, as they all rely on a priori knowledge and training. This motivates our exploration of TTA as a more suitable solution.

## 3 PROBLEM SETTING

Following (Xiao & Snoek, 2024; Liang et al., 2024), we assume access to a regressor $f_\theta : \mathcal{X} \to \mathbb{R}^d$ pre-trained on *source* samples $(\mathbf{x}_i, \mathbf{y}_i)_{i=1}^{N^{\mathrm{src}}} \in \mathcal{X} \times \mathbb{R}^d$ drawn from a source distribution $P^{\mathrm{src}}$, e.g., $f_\theta = g \circ \phi$ in Fig. 1. We also assume access to some real matrix-valued source statistics.

The goal is, for any new *unlabeled* sample $(\mathbf{x}_i^{\mathrm{tgt}})_{i=1}^{N^{\mathrm{tgt}}}$ drawn from the input marginal of a *target* distribution $P^{\mathrm{tgt}} \neq P^{\mathrm{src}}$, to find $\theta$ which minimizes the risk

$$\mathcal{R}(f_\theta) = \frac{1}{N^{\mathrm{tgt}}} \sum_{i=1}^{N^{\mathrm{tgt}}} \left\| f_\theta(\mathbf{x}_i^{\mathrm{tgt}}) - \mathbf{y}_i^{\mathrm{tgt}} \right\|_2^2.$$

Note that we have no access to any target labels $(\mathbf{y}_i^{\mathrm{tgt}})_{i=1}^{N^{\mathrm{tgt}}}$. Therefore, the target risk $\mathcal{R}(f_\theta)$ cannot be evaluated directly and importance weighting (Shimodaira, 2000) cannot be used without further modification.

We study the problem above using SIMSHIFT (Setinek et al., 2025) and EngiBench (Felten et al., 2025) as datasets. SIMSHIFT is designed to evaluate how surrogate models adapt to distribution shifts on real-world industrial simulation tasks, while EngiBench is a collection of design optimization datasets, optimizers, and simulators to evaluate designs. In both benchmarks, the inputs $\mathbf{x}$ represent parameters like geometry, material properties, desired or operating conditions. The "labels" $\mathbf{y}$ correspond to high-dimensional fields such as stresses or deformation for SIMSHIFT, and material density of the generated design for EngiBench.

In both cases the target distribution originates from unseen parameter configurations and the goal is to predict the corresponding fields. While SIMSHIFT formulates the problem as a regression task with neural operators (Kovachki et al., 2021), EngiBench treats it as an inverse problem solved by generative models. The diversity of tasks and training techniques in the two frameworks reinforces the model-agnostic aspect of our method.

## 4 METHOD

In the following section we describe the process to obtain SATTS. First, we define the "inner" component of TTA, which involves representation alignment for simulations (ATTS). Next, we describe how to stabilize TTA using a "source-less" model selection algorithm (SATTS).

### 4.1 REPRESENTATION ALIGNMENT WITH ATTS

A common approach in TTA classification is to use feature alignment to reduce the discrepancy between source and target distributions (Ishii & Sugiyama, 2021; Adachi et al., 2025). Following

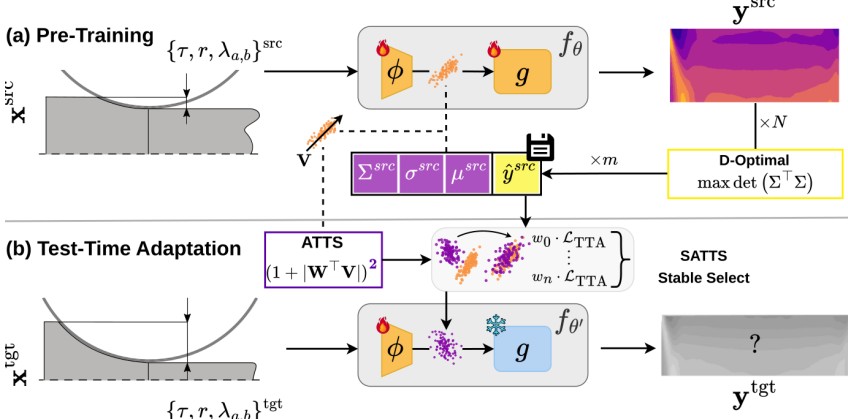

Figure 1: Overview of training and test-time adaptation, using *hot rolling* from SIMSHIFT as an example. (a) Pre-training on the source domain using input parameters: thickness ($\tau$), post-rolling reduction ($r$), and temperature coefficients ($\lambda_a$, $\lambda_b$). The *representation learner* $\phi$ and the *predictor g* are optimized jointly. (b) Test-time adaptation on the target domain, where only the input parameters are available. Here $\phi$ is adapted and $g$ is frozen. Statistical source features enable the selection of the best-performing model.

this principle, we design our regression network with two connected components: a *representation learner* $\phi$ that produces intermediate features, and a *predictor g* that maps these features to outputs. Adaptations occur in the representation stage, while the predictor remains unchanged. Fig. 1 sketches this process using hot rolling as an example, and distinguishes between (a) pretraining and (b) TTA with ATTS, and stabilization with model selection for SATTS.

The idea behind Adaptation at Test-Time for Simulation (ATTS) is to adjust the features $\mathbf{z} := \phi(\mathbf{x})$, $\mathbf{z} \in \mathbb{R}^C$ such that the target features are similarly distributed as the source features. During TTA, target batch statistics are computed on-the-fly, while source statistics $\Sigma^{\text{src}}, \mu^{\text{src}}, \sigma^{\text{src}}$ (pre-computed and stored after training) are used to align the source and target feature distributions by minimizing the Kullback-Leibler divergence (see Appendix A).

ATTS operates on the full feature space and assigns importance through *dimension weighting*, implemented via exponentiation of the weighting function. This avoids the need for manually restricting the representation to a pre-selected subset of principal components. Leading to the following formulation:

$$\boldsymbol{\alpha} = (1 + |\mathbf{W}^\top \mathbf{V}^{\text{src}}|)^2, \tag{1}$$

where $\mathbf{W} \in \mathbb{R}^{K \times C}$ are weights from the first layer of the predictor $g$ (for a $C$-dimensional $\mathbf{z}$) and $\mathbf{V}^{\text{src}} \in \mathbb{R}^{K \times K}$ is the principal component basis of the source features. Representation alignment is applied to the target features $\mathbf{z}^{\text{tgt}} := \phi(\mathbf{x}^{\text{tgt}})$. Each channel $\mathbf{z}_c^{\text{tgt}}$ is projected onto the source basis $\mathbf{V}^{\text{src}}$, reweighted by the corresponding factor $\boldsymbol{\alpha}_c$ with $c \in [0, C-1]$ as

$$\tilde{\mathbf{z}}_c^{\text{tgt}} = \left(\mathbf{z}_c^{\text{tgt}} - \mu^{\text{src}}\right) \mathbf{V}^{\text{src}} \boldsymbol{\alpha}_c, \tag{2}$$

where $\boldsymbol{\alpha}_c$ is the $c$-th row of $\boldsymbol{\alpha}$ (channel-wise). By using Eqs. (1) and (2) to do feature selection, ATTS can preserve a rich set of features, improving adaptation under distribution shift for simulation datasets. Furthermore, since our targets are vector-valued, we compute the Kullback-Leibler divergence independently for each channel, rather than combining them into a single space.

## 4.2 STABLE ADAPTATION

SATTS extends the representation alignment method proposed in Section 4.1 with online model selection in order to ensure stable results. However, a general rule in Test-Time Adaptation (TTA) is that source data is not accessible after training and only *statistics* can be stored. Most application

strategies, such as accuracy-on-the-line (Miller et al., 2021) or density ratio estimation Sugiyama et al. (2007); You et al. (2019); Dinu et al. (2023), rely on access to source performance (or full source datasets), making model selection in TTA non trivial. To address this point, we approximate source performance through the optimal selection of a fixed number of representative source feature statistics. Since the size of this spanning set is constant regardless of dataset size, it can be confidently treated as an additional source statistic.

Given an information matrix $X$, *D-Optimality* picks a spanning set of indices $S \subset \mathbb{Z}, \quad |S| = m$, selecting vectors that maximize the determinant of $X$, $\max \det(X^\top X)$, maximizing the volume spanned by the chosen vectors (Atkinson & Donev, 1992). Originating from experimental design, this criterion identifies the most informative subset of $m$ points, ensuring that they preserve the maximal variance information from the original feature space. However, the D-optimal criterion can grow intractable for large datasets, as it requires the full covariance. Instead, we approach this by using QR pivoting on whitened principal components as an efficient approximation (Golub & Van Loan, 2013). The algorithm employs a PCA dimensionality reduction with QR decomposition, resulting in a quasi D-optimal choice of indices $S$. We showcase it in Algorithm 1. Source performance can be estimated by evaluating models on the optimal set, enabling model selection in a *"source-less"* setting. The indices $S$ returned by Algorithm 1 are used to select the spanning set of source fields $\hat{y}^{src}$ (yellow box in Fig. 1), stored during pretraining.

Model selection is performed via line search over the TTA learning rate hyperparameter in the range $[1 \times 10^{-5}, 1 \times 10^{-2}]$. After each adaptation step, performance is evaluated on the D-Optimally selected source representatives, and the search stops once further decreases no longer improve the objective. The best learning rate is then selected for the final evaluation of the TTA algorithm.

---

**Algorithm 1** Quasi D-optimal spanning set selection via PCA and QR pivoting

---

**Require:** Dataset $\mathbf{D} \in \mathbb{R}^{N \times d}$, eigendecomposition $(\boldsymbol{\lambda}, \mathbf{V})$, variance threshold $\tau$, desired samples $m$
**Ensure:** Selected source datast indices $S \subseteq \{1, \dots, N\}$
1: $\mathbf{D} \leftarrow \mathbf{D} - \text{mean}(\mathbf{D})$
2: $r \leftarrow \text{select\_components}(\boldsymbol{\lambda}, \tau)$               ▷ by keeping $\tau\%$ of the variance
3: $\mathbf{Z} \leftarrow \mathbf{D}\mathbf{V}_{:,1:r}\mathbf{\Lambda}_r^{-1/2}$
4: $\mathbf{Q}, \mathbf{R}, \text{piv} \leftarrow \text{QR}(\mathbf{Z}^T)$
5: $S \leftarrow \text{piv}[1:m]$
6: **return** $S$

---

## 5 EXPERIMENTS

In our experiments, we focus on two specific datasets that illustrate complex industrial use cases where neural surrogates approximate costly numerical simulations. Both datasets exhibit distribution shifts resulting from variations in input parameters, such as initial conditions, geometries or material types. In the following subsections, we will highlight the performance and stability of our method, first on the SIMSHIFT benchmark (Setinek et al., 2025) and then on the EngiBench dataset (Felten et al., 2025). In all experiments, the parameters of the quasi Algorithm 1 are fixed to $m = 5$ indices and a threshold of $\tau = 0.95\%$.

### 5.1 NEURAL SURROGATES FOR SIMULATION: SIMSHIFT

To investigate the performance on industrial simulation tasks, we use the SIMSHIFT datasets (Setinek et al., 2025), which span four distinct industrial simulation settings: *hot rolling*, *sheet metal forming*, *electric motor*, and *heatsink design*. All datasets have explicit source and target domain splits, dependent on the parameters such as initial conditions, material or geometry specifications that were used to generate the samples. Shifts happen in parametric space, as opposed to unstructured variations occurring in images. We perform all our experiments using the medium difficulty domain shift setup for all datasets. For a detailed description of the datasets, their creation, and the defined distribution shifts, we refer the reader to the SIMSHIFT publication (Setinek et al., 2025).

Table 1 summarizes the results across all datasets, comparing our method against SSA, *"unregularized"* pre-trained predictions (*"Source"*), and Unsupervised Domain Adaptation (UDA) applied to the

Table 1: Comparison of current baselines with TTA methods for all simulation datasets. Results are averaged across 20 TTA runs, over 4 models (80 seeds in total) with standard deviation reported. Reported RMSE is normalized over all fields.

(a) Rolling

| Model | RMSE ($\downarrow$) | MAE ($\downarrow$) | $R^2$ ($\uparrow$) |
|---|---|---|---|
| Source | $0.540_{\pm0.068}$ | $0.468_{\pm0.060}$ | $0.789_{\pm0.077}$ |
| UDA | $0.432_{\pm0.020}$ | $0.378_{\pm0.073}$ | $0.813_{\pm0.088}$ |
| SSA | $0.559_{\pm0.073}$ | $0.488_{\pm0.067}$ | $0.789_{\pm0.078}$ |
| SATTS | $\mathbf{0.538_{\pm0.070}}$ | $\mathbf{0.467_{\pm0.062}}$ | $\mathbf{0.789_{\pm0.079}}$ |

(b) Motor

| Model | RMSE ($\downarrow$) | MAE ($\downarrow$) | $R^2$ ($\uparrow$) |
|---|---|---|---|
| Source | $0.109_{\pm0.002}$ | $0.058_{\pm0.001}$ | $0.989_{\pm0.000}$ |
| UDA | $0.099_{\pm0.002}$ | $0.053_{\pm0.001}$ | $0.989_{\pm0.001}$ |
| SSA | $0.110_{\pm0.003}$ | $0.058_{\pm0.001}$ | $0.988_{\pm0.001}$ |
| SATTS | $\mathbf{0.109_{\pm0.003}}$ | $\mathbf{0.058_{\pm0.001}}$ | $\mathbf{0.989_{\pm0.000}}$ |

(c) Forming

| Model | RMSE ($\downarrow$) | MAE ($\downarrow$) | $R^2$ ($\uparrow$) |
|---|---|---|---|
| Source | $0.137_{\pm0.008}$ | $0.052_{\pm0.003}$ | $0.984_{\pm0.002}$ |
| UDA | $0.109_{\pm0.004}$ | $0.043_{\pm0.002}$ | $0.985_{\pm0.001}$ |
| SSA | $0.138_{\pm0.011}$ | $0.052_{\pm0.004}$ | $0.983_{\pm0.002}$ |
| SATTS | $\mathbf{0.136_{\pm0.008}}$ | $\mathbf{0.052_{\pm0.003}}$ | $\mathbf{0.984_{\pm0.002}}$ |

(d) Heatsink

| Model | RMSE ($\downarrow$) | MAE ($\downarrow$) | $R^2$ ($\uparrow$) |
|---|---|---|---|
| Source | $0.634_{\pm0.012}$ | $0.424_{\pm0.004}$ | $0.484_{\pm0.027}$ |
| UDA | $0.577_{\pm0.005}$ | $0.374_{\pm0.001}$ | $0.553_{\pm0.002}$ |
| SSA | $0.632_{\pm0.014}$ | $0.424_{\pm0.003}$ | $0.484_{\pm0.026}$ |
| SATTS | $\mathbf{0.631_{\pm0.014}}$ | $\mathbf{0.423_{\pm0.003}}$ | $\mathbf{0.485_{\pm0.026}}$ |

Table 2: Comparison of current baselines with TTA methods for design optimization datasets. Results are averaged across 20 TTA runs, over 4 models (80 seeds in total) with standard deviation reported.

(a) Beams2D

| Model | COMP ($\downarrow$) | MAE ($\downarrow$) | MMD ($\downarrow$) |
|---|---|---|---|
| Source | $123.7_{\pm17.854}$ | $\mathbf{0.026_{\pm0.004}}$ | $\mathbf{0.052_{\pm0.002}}$ |
| SSA | $119.4_{\pm4.586}$ | $0.040_{\pm0.005}$ | $0.062_{\pm0.003}$ |
| SATTS | $\mathbf{118.8_{\pm12.409}}$ | $0.027_{\pm0.004}$ | $0.053_{\pm0.002}$ |

(b) HeatConduction2D

| Model | COMP ($10^{-3}$) | MAE ($\downarrow$) | MMD ($\downarrow$) |
|---|---|---|---|
| Source | $0.577_{\pm0.561}$ | $0.336_{\pm0.057}$ | $0.095_{\pm0.000}$ |
| SSA | $0.712_{\pm0.615}$ | $0.349_{\pm0.057}$ | $0.095_{\pm0.000}$ |
| SATTS | $\mathbf{0.537_{\pm0.491}}$ | $\mathbf{0.334_{\pm0.057}}$ | $0.095_{\pm0.000}$ |

pre-trained model. For implementation details see Appendix B. SATTS consistently outperforms SSA, establishing a new baseline for test-time adaptation in neural surrogate regression. While improvements over the source model may be marginal in some cases, SSA can destabilize the pre-trained model, whereas our approach does not degrade performance. Moreover, when using UDA as a lower bound, our method reduces the gap without fully closing it, leaving room for future improvement. However, directly comparing them is unfair for TTA, as UDA has access to the target input distribution during pretraining and takes significantly more time at pretraining. Since UDA also requires unsupervised model selection and a hyperparameter sweep over the loss-balancing coefficient, its pretraining pipeline took nine times longer than standard pretraining on the SIMSHIFT datasets.

## 5.2 GENERATIVE DESIGN OPTIMIZATION: ENGIBENCH

We evaluate on two EngiBench design–optimization tasks: *structural beam bending* and *2D heat conduction*. By default, these datasets do not include predefined source and target domains. We therefore define them following the approach in Setinek et al. (2025): we train models on the full datasets and subsequently analyze the t-SNE visualizations of the latent feature spaces as the input conditions are varied. Datasets are then partitioned into source and target domains based on the parameters that dominate the latent space variation. A detailed analysis of this procedure and corresponding visualizations can be found in Appendix D.

We report Mean Absolute Error (MAE), the Maximum Mean Discrepancy (MMD), and Compliance (COMP), a dataset specific objective value calculated with a Finite Element Method (FEM) solver. For Beams2D, compliance is the inverse of stiffness whereas for HeatConduction2D it is the thermal compliance coefficient. In Table 2, we compare our method against the *"unregularized"* pre-trained model (*"Source"*), and SSA. Across both tasks, our approach typically matches or reduces errors relative to the unregularized model. Compared to our method, SSA shows unstable behavior on certain metrics, sometimes even deteriorating performance substantially. Such behavior is highly undesirable in TTA deployments and underlines the strong suit of our approach: its stability.

| Method | $K$ | RMSE $\downarrow$ | MAE $\downarrow$ |
|--------|-----|---------|---------|
| SSA | 10 | $0.631_{\pm 0.128}$ | $0.541_{\pm 0.107}$ |
|  | 20 | $0.559_{\pm 0.072}$ | $0.488_{\pm 0.067}$ |
|  | 30 | $0.573_{\pm 0.086}$ | $0.504_{\pm 0.077}$ |
|  | 50 | $0.626_{\pm 0.098}$ | $0.553_{\pm 0.088}$ |
|  | 100 | $0.644_{\pm 0.106}$ | $0.569_{\pm 0.094}$ |
| ATTS | All | $0.543_{\pm 0.092}$ | $0.473_{\pm 0.058}$ |
| SATTS | All | $\mathbf{0.538}_{\pm 0.070}$ | $\mathbf{0.467}_{\pm 0.062}$ |

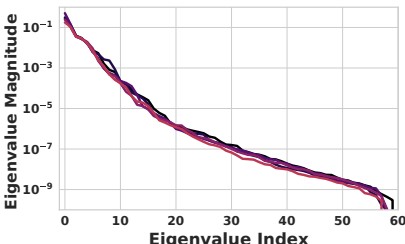

(a) Comparison of SATTS with SSA for different choices of $k$.  (b) Eigenvalue analysis.

Figure 2: Ablations with SSA (Adachi et al., 2025) for the hot rolling dataset. (a) table with quantitative comparison, (b) eigenvalues analysis for different trained models, highlighting the fast decay. $\sim 60\%$ of the energy is on the first eigenvalues, favoring compact representations.

## 5.3 ABLATIONS

**ATTS feature weighting mechanism.** Different datasets can experience varying degrees of improvement from TTA, highlighting the unique complexities of each problem. By analyzing the eigenvalue distribution (Fig. 2b for hot rolling), we observe that improvements correlate with the explanatory power of the leading eigenvalues: datasets where a few components capture most of the variance exhibit stronger adaptation gains. In contrast, in the motor dataset, variance is spread across many eigenvalues, indicating a higher-dimensional problem structure and limiting the effectiveness of current TTA methods. See Appendix C for the full eigenvalue analysis.

To illustrate this point, we ablate the impact of the number of dimensions $K$ of the feature subspace for the standard SSA algorithm in Fig. 2. Originally, $K$ has to be chosen manually from the eigenvalue spectrum of the covariance matrix, requiring expert interaction. While the variance in the hot rolling datasets decays sharply after the first ten directions (Fig. 2b), several low-variance components remain correlated with regression targets. SSA results in Fig. 2a suggest that a handpicked $K$ might not be optimal, and strict truncation can discard relevant information. For multivariate regression problems, feature selection is thus critical. Importantly, this choice is absent in ATTS, simplifying the tuning and yielding superior results. Fig. 2a shows that across different $K$-values, our method always outperforms SSA, irrespective to the chosen subspace size.

**Compute.** Our method stabilizes and often improves inference without any a priori knowledge of the test-time distribution, at a small computational overhead. Concretely, we add at most $L$ forward and backward passes, where $L$ represents maximum number of candidate learning rates used by the line search. In particular, for a "source" model with complexity $O(N)$, with $N$ samples, TTA has a complexity of $O(2NL)$ at worst, with $L << N$ sequential evaluations for the line search and 2 for the two forward passes required by training and model selection. Additionally, unsupervised model selection adds a small cost, which is proportional to the speed of the chosen criterion. While the results produced by UDA serve as a lower bound for our method, comparing compute is not ideal since it requires different pretraining with substantially increased costs: It uses source and target data, therefore being computationally more expensive and having the disadvantage of requiring access to the test-time distribution a priori.

## 6 CONCLUSION AND FUTURE WORK

In this work, we make the initial step towards accurate test-time adaptation methods for neural surrogates, and in general for high-dimensional multivariate regression. Our findings show that the proposed adjustments enable TTA to yield zero-shot improvements at negligible computational cost. By leveraging latent covariance structures for distribution alignment and stabilizing it with automated parameter selection, our approach ensures stable and accurate test-time adaptation in high-dimensional multivariate regression.

In addition to the near-zero cost gains, this line of research is particularly timely due to evolving compliance requirements. Article 15 of the EU Artificial Intelligence Act states that high-risk AI systems need to ensure appropriate levels of accuracy and robustness (EUA, 2024). Should neural surrogates be deployed in safety-critical domains, such as accelerating structural design in the automotive industry, accurate and reliable predictions become indispensable.

However, performance improvements are generally minor, and the lower bounds established by UDA indicates that additional gains remain attainable. This points to the potential for a new class of TTA algorithms, specifically developed for physics simulation data. We foresee two paths to achieve *"physics-driven"* TTA that are to be explored: (i) use physics-informed constraints and priors (Raissi et al., 2019; Cai et al., 2021), ad-hoc and calibrated on the test case, to augment the expressiveness of the limited test labels, and (ii) incorporate uncertainty quantification to localize failure regions in the fields where adaptation is necessary.

## REPRODUCIBILITY STATEMENT

We provide detailed descriptions of our approach SATTS, by describing the models, algorithm, and experimental setup in the main paper, with additional implementation details in the appendix. All datasets used in this work are publicly available, and we include the details on the data splits for EngiBench (Felten et al., 2025) in the Appendix D. Upon publication, we will release the code for the provided method on GitHub.

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

## LLM Usage Disclosure

In general, LLM tools were used to refine writing in multiple parts of the paper, such as introduction, method and experiment sections (GPT-5). Claude3.5 and GPT-5 were additionally make visualizations prettier, speed up the development of plotting functions, and dump results neatly into tables. Beyond that, they were also used to a lesser degree in other parts of the code (Cursor). The literature review was done manually, with web searches and interaction from experts in each field, as language models would end up in unsatisfactory results most of the times. Overall, AI assistants were strictly used for editing and never in ideation beyond understanding other work.

## A    Representation Alignment and TTA Setup

For the TTA experiments, we utilize validation source data to compute statistical information. Within this step, we further collect the indices of the stored source samples of the validation source data based on Algorithm 1. Model updates are carried out based on the target test datasets, while evaluation is conducted on both the target data and the sampled source data. The TTA updates are constrained to the number of batches within each dataset.

In our specific setup, task-dependent parameters, such as thickness or temperature, are encoded through a conditioner network. The conditioner is divided into two components: a main body and a final linear layer. We extract features from the main body's output and define the split between representation learner and predictor at this point—the conditioner serves as the representation learner, while, for example, the Transolver acts as the predictor. This choice reflects the observation that most task-related parameter shifts occur within the conditioning network.

At test time, the precomputed source statistics are used to project the target features into the significant subspace. From the projected target features, their mean and variance $(\tilde{\mu}_k^{\text{tgt}}, \tilde{\sigma}_k^{\text{tgt2}})$ are calculated and aligned with the corresponding source statistics $(0, \lambda_k^{\text{src}})$. We restrict adaptation the layer normalization (Ba et al., 2016) parameters of the conditioner, keeping all other model parameters fixed. For training ATTS and SATTS we use the weighted symmetric Kullback-Leibler divergence to account for Eq. (2) by removing $\boldsymbol{\alpha}_k$

$$\mathcal{L}_{\text{TTA}} = \frac{1}{2} \sum_{k=1}^{K} \left( \frac{(\tilde{\mu}_k^{\text{tgt}})^2 + \lambda_k^{\text{src}}}{\tilde{\sigma}_k^{\text{tgt2}}} + \frac{(\tilde{\mu}_k^{\text{tgt}})^2 + \tilde{\sigma}_k^{\text{tgt2}}}{\lambda_k^{\text{src}}} - 2 \right). \tag{3}$$

All experiments are conducted with a fixed batch size of 32. To ensure robustness, we repeat each experiment with 20 different random seeds per model for the SIMSHIFT benchmark and 10 seeds for the structural beam bending and two seeds for the 2D heat conduction dataset. The varying seeds originate from the number of data samples available in each dataset. Since the 2D heat conduction dataset is limited in size and it is effectively contained in a single test-time batch, increasing the number of seeds did not influence the performance of the TTA algorithm. This is particularly important since layer normalization is updated online, after every batch.

**Model selection.** As outlined in 4.2, model selection is executed after TTA with a chosen learning rate. The evaluation criterion for terminating the search over learning rates ($lr$) is based on performance measured on the source statistics. We employ the Root Mean Squared Error (RMSE) for the SIMSHIFT dataset and COMP for EngiBench. We set the hyperparameter search for the learning rates to $[0.05, 0.01, 0.005, 0.001, 0.0005, 0.0001]$.

For comparison, we also report the best performing UDA algorithm as a lower bound in the SIMSHIFT datasets. These models are trained according to the procedure outlined in SIMSHIFT (Setinek et al., 2025). We run the DeepCoral algorithm (Sun & Saenko, 2016) with the provided $\lambda$ ranges. After applying selection and emsembling strategies on top of the UDA algorithm, we showcase the best-performing model for each dataset. It is important to note that the UDA training process requires significantly more compute budget than the TTA approach: instead of requiring a single pre-trained model, UDA model selection relies on multiple models for robustness, each trained independently with different $\lambda$ values.

# B   EXPERIMENTAL SETUP

In the following paragraphs, we detail the experimental setup, including the selected models and our training and testing strategy.

## B.1   MODEL ARCHITECTURES & PRETRAINING

We employ different model architectures to evaluate our TTA method. The models are based on the architectures provided in the benchmark datasets Setinek et al. (2025) and Felten et al. (2025), implemented in PyTorch, and designed for conditional regression or optimization tasks. Node coordinates are provided as inputs and embedded using sinusoidal positional encodings. Conditioning is applied through a dedicated network that processes the simulation input parameters.

**Conditioning Network.** The conditioner maps simulation parameters into a latent representation of dimension 8. It consists of a sinusoidal encoding, followed by a small MLP, which includes two LayerNorms to stabilize training.

**Transolver.** The Transolver architecture (Wu et al., 2024a) starts by encoding node coordinates using sinusoidal position embeddings, followed by an MLP that produces initial feature vectors. A learned mapping then assigns each node to a slice, enabling attention operations both within slices and between them. The processed features are passed through an MLP readout to generate the final field outputs. Two conditioning mechanisms are available: concatenating the conditioning vector with input features or applying it via DiT-based modulation across the network. Conditioning is done with the dit-based modulation (Peebles & Xie, 2023). Where a latent dimension of 128, a slice base of 32, and four attention layers are used. This results in a model with 0.57M parameters. We additionally employ a larger model with 56, 128, and 8 layers for the more complex dataset, leading to 4.07M parameters.

**Diffusion Model.** As a diffusion model, we employ a conditional U-Net (Ronneberger et al., 2015) from Hugging Face's `diffusers` library[1].

The model works as a denoiser, taking a noisy field and a conditioning vector from the conditioning network described above and producing a noise prediction. We summarize all hyperparameters of our diffusion model in Table 3. To train the model, we use the standard Denoising Diffusion Probabilistic Models (DDPM) objective of noise prediction ("$\epsilon$-prediction") with 100 diffusion steps and a `squaredcos_cap_v2` beta scheduler.

**Pretraining setup.** All unregularized baseline (*"Source"*) models are pretrained using the following setup: We use an initial learning rate of $10^3$ with a cosine decay scheduler and weight decay of $10^{-5}$. Training runs for up to 500, 1500, and 3000 epochs on *Beams2D*, *HeatConduction2D*, and *SIMSHIFT*, respectively, with early stopping if the validation loss does not improve for 500 epochs. We enable gradient clipping and maintain an Exponential Moving Average (EMA) of the model parameters with decay 0.95. Automatic Mixed Precision (AMP) ius enabled only for the large scale *heatsink* dataset; for all others we train in `float32`. Batch size is 64 for EngiBench baselines and 16 for SIMSHIFT baselines.

Table 3: Hyperparameters for our conditional diffusion U-Net. This setup leads to a model size of 17.5M parameters.

| Hyperparameter | Value | HF Class Argument Name |
|---|---|---|
| Block channels (low→high) | $[32, 64, 128, 256]$ | `block_out_channels` |
| Layers per block | 2 | `layers_per_block` |
| Transformer layers / block | 1 | `transformer_layers_per_block` |
| Cross-attention dim | 64 | `cross_attention_dim` |
| Only cross-attention | True | `only_cross_attention=True` |
| Normalization groups | 16 | `norm_num_groups` |
| Activation | SiLU | `act_fn` |

---

[1] `UNet2DConditionModel`

## C  ADDITIONAL RESULTS

To investigate the difference in effectiveness of TTA algorithms to different datasets, we analyze the features extracted by the representation learner. Specifically, the eigenvalue spectra of the corresponding covariance matrices are compared for each dataset in SIMSHIFT (Setinek et al., 2025). In Fig. 3, the eigenvalues for three out of four datasets (hot rolling, sheet metal forming, and heatsink design) show a similar decay: the first five eigenvalues already capture up to ~60% of the total variance. This is also confirmed by Table 4. In contrast, the electric motor dataset exhibits a much slower decay, suggesting that the variance is distributed across a larger number of components. This implies that electric motor requires a higher-dimensional representation to preserve the same level of information, whereas the other datasets are more efficiently represented in a low-dimensional manifold.

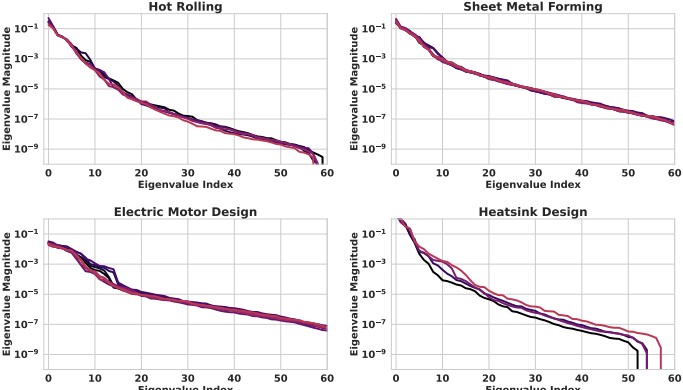

Figure 3: Eigenvalues analysis of *hot rolling*, *sheet metal forming*, *electric motor* and *heatsink design*, expressing the diverging decay throughout the datasets.

Table 4: Percentage of variance explained by the top 10 eigenvalues for each dataset.

| Dataset | $\lambda_0$ | $\lambda_1$ | $\lambda_2$ | $\lambda_3$ | $\lambda_4$ | $\lambda_5$ | $\lambda_6$ | $\lambda_7$ | $\lambda_8$ | $\lambda_9$ |
|---------|------|------|------|------|------|------|------|------|------|------|
| Rolling | 57.9% | 23.6% | 7.1% | 5.4% | 3.4% | 1.4% | 0.7% | 0.3% | 0.2% | 0.1% |
| Forming | 47.6% | 18.3% | 14.2% | 8.6% | 5.6% | 3.0% | 1.3% | 0.8% | 0.4% | 0.2% |
| Motor | 25.0% | 19.5% | 15.2% | 13.2% | 10.4% | 7.6% | 4.3% | 2.7% | 1.3% | 0.7% |
| Heatsink | 63.9% | 22.3% | 8.5% | 4.2% | 0.5% | 0.2% | 0.1% | 0.1% | 0.0% | 0.0% |

## D  DISTRIBUTION SHIFTS FOR ENGIBENCH

Figs. 4 to 5 show t-SNE visualizations of the conditioning-networks' latent spaces for models trained across the full range condition variables. For *structural beam bending* (Fig. 4), `volfrac` and `rmin` cause the clearest structure in latent space. We therefore chose to split the source and target domain depending on `rmin`. For *2D heat conduction* (Fig. 5), `volume` and `length` exhibit comparable influence on the latent space distribution. Following the same protocol, split along `volume`. The resulting source and target ranges and sizes for both datasets can be found in Table 5.

Table 5: Defined distribution shifts (source and target domains) for each dataset.

| Dataset | Parameter | Description | Source range (no. samples) | Target range (no. samples) |
|---------|-----------|-------------|----------------------------|----------------------------|
| Beams2D | rmin | Minimum feature length of beam members. | [1.5, 3.25) (3087) | [3.25, 4] (353) |
| HeatConduction2D | volume | Volume limits on the material distributions. | [0.3, 0.465) (231) | [0.465, 0.6] (39) |

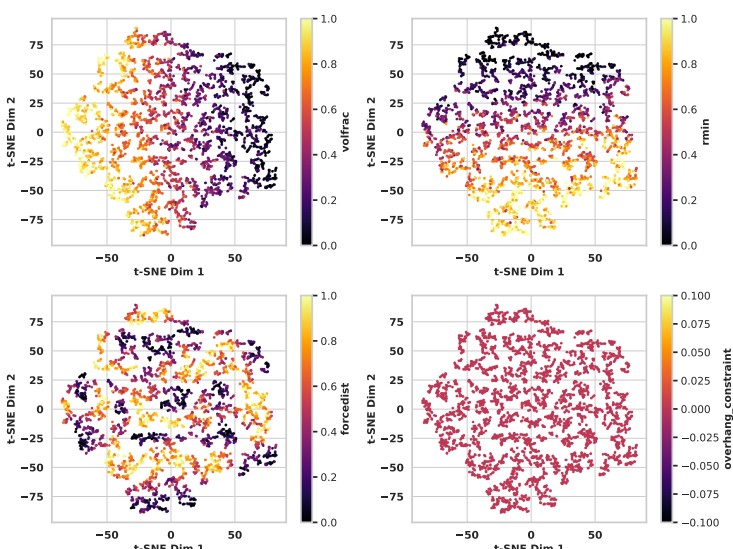

Figure 4: t-SNE visualization of the conditioner's latent space on the *structural beam bending* dataset. While `overhang_constraint` and `forcedist` are either constant or exhibit almost a uniform distribution, `volfrac` and `rmin` exhibit a clear structure.

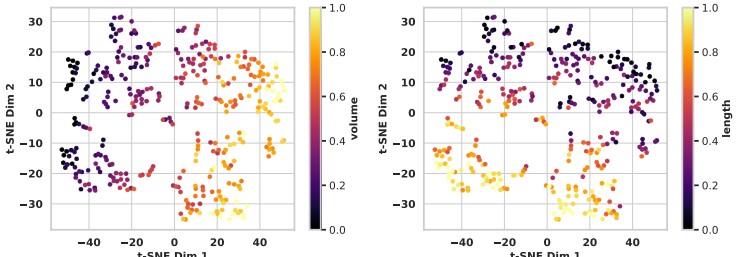

Figure 5: t-SNE visualization of the conditioner's latent space on the *structural beam bending* dataset. Both conditions (`volume` and `lentgh`) exhibit a clear structure in the latent space.

