# OpenReview forum: "Stabilized Test-Time Adaptation of Neural Surrogates in Simulation"
_ICLR.cc/2026/Conference — ICLR 2026 Conference Withdrawn Submission_

### Official Review · Reviewer_eLsg · 2025-10-19

**Soundness:** 2
**Presentation:** 3
**Contribution:** 2
**Rating:** 2
**Confidence:** 3

**Summary:**

This paper adapts test-time adaptation (TTA) high-dimensional neural surrogates. The core is a feature-alignment procedure (ATTS) that weights full latent channels by their importance and then aligns target batch statistics to stored source statistics. This is combineed with a "source-less" model-selection routine (quasi D-optimal selection + line-search over learning rates) to stabilize adaptation (SATTS). The paper evaluates SATTS on two benchmarks (SIMSHIFT, EngiBench).

**Strengths:**

1. The paper identifies an underexplored application area (TTA for neural surrogates / high-dimensional regression) and motivates why deployment constraints make their proposed method attractive.
2. To my knowledge, the methods introduced by the authors are intuitive and relevant for the proposed problem setting.

**Weaknesses:**

3. Is the source distribution necessarily normal? For more complex distributions, the source statistics that are collected in ATTS may not be sufficient to adequately capture the source distribution.
4. The authors state that all experiments were run using the medium difficulty domain shift setup in line 265. It would be informative to include ablations of the domain shift difficulty (especially the "hard" setting but also "easy").
5. The results in Table 1 seem to have very large standard deviations - a back-of-the-envelope estimation of the 95% confidence intervals of the RMSE values suggest that there is rarely (if ever) a meaningful improvement between SATTS and SSA, for instance. In contrast, there seems to be a statistically significant difference between UDA and SAATTS in Table 1.
6. I think it would also be important to ablate the batch size for SATTS training, as it affects Equation (3) and training dynamics in Appendix A.
7. Similarly, how would ablating the $\tau$ variance threshold in Algorithm 1 for SATTS affect performance?
8. I am not as familiar with recent work on TTA, but would prior methods such as [TENT](https://arxiv.org/abs/2006.10726) and/or [conjugate psuedo-label TTA](https://arxiv.org/abs/2207.09640) be relevant to compare against? I also wonder how an oracle positive control method might perform - i.e., a model trained on the target domain with the actual labels (that aren't available to other methods); this might help understand how far the methods are from the optimal supervised setting. Is it also possible to derive a small number of pseudo-labels for the target samples, and use those to update just the Layer Norms as a sort of "minimalist", simplistic baseline?
9. The novelty of the proposed method seems to be rather minimal - my impression is that the method combines standard TTA techniques with a PCA-based "importance score" to determine which dimensions to focus on for aligning. I am cognizant that algorithmic novelty is by no means required for ICLR, but am not sure if the empirical contributions otherwise are sufficient given the concerns raised above.

Minor Comments:
  - "datast" is missing an "e" in line 241

**Questions:**

10. What is the intuition behind Equation (1)?
11. I understand that Algorithm (1) is used to select the $m$ "most informative" $y$ observations from the source dataset. Could you also do something similar to select the "most informative" $x$ observations as the "statistics" too, and use them to better capture the properties of the source distribution?
12. What is the main takeaway of Figures 4 and 5?

---

### Official Review · Reviewer_bPfU · 2025-10-31

**Soundness:** 2
**Presentation:** 1
**Contribution:** 2
**Rating:** 2
**Confidence:** 3

**Summary:**

This paper introduces SATTS, a method aiming to adapt pre-trained neural surrogate models to target domains without access to labeled data. SATTS consists of two parts: (1) ATTS, a feature-alignment stage that uses stored source statistics and a KL divergence to align target latent representations, and (2) a stabilized model-selection step that performs a line search over the learning rate for adaptation, guided by a small D-optimal subset of stored source samples. Experiments are conducted on SIMSHIFT and ENGIBENCH datasets, comparing SATTS to UDA and SSA, and showing marginal performance improvements in some cases.

**Strengths:**

- The paper addresses the practical challenge of enhancing the performance of neural surrogates during deployment, particularly in cases where there is a distribution shift between the training and test data.
- The proposal method aims to improve adaptation robustness by integrating feature alignment with a small-scale model-selection mechanism.

**Weaknesses:**

- The proposed SATTS combines two well-known ideas, feature alignment (via a KL divergence) and hyperparameter line search for learning rate tuning, neither of which is novel. The core of SATTS (performing line search over learning rates) is essentially a hyperparameter selection procedure, not a new algorithmic principle. The claimed “stabilization” is achieved by tuning, not by introducing a fundamentally robust adaptation mechanism.
- Only two baselines (UDA and SSA) are compared, which is far from sufficient to demonstrate the strength of the proposed approach. Furthermore, UDA consistently outperforms SATTS (see Table 1), which undermines the claim that SATTS is more effective, especially since the authors admit UDA’s pretraining cost is higher, but fail to compare inference-time performance, which might make UDA still more desirable overall. The EngiBench experiment omits comparison with UDA entirely, which weakens the evaluation and makes the results incomplete.
- The paper's writing requires significant improvement to enhance its quality.

**Questions:**

- How is the KL divergence term defined exactly in your method? It would be helpful to include its explicit mathematical form in the main text (Line 192), especially since there appears to be sufficient space remaining in the paper. In its current form, the method section is not entirely self-contained and could benefit from a more careful and precise presentation.
- Why are there only two baselines compared to SATTS? In particular, why was UDA not included in the EngiBench experiments? Moreover, since UDA consistently outperforms SATTS, it is unclear why practitioners should prefer SATTS if inference costs are comparable.
- How sensitive is the method to the range of learning rates? Describing this simple line search as a form of “model selection” seems overstated, as it is effectively a hyperparameter tuning procedure rather than a genuine model-selection strategy.
- Minor points:
     - Line 95: Typo (“approximatethe”).
     - Fig. 1: Lacks explanation of important symbols ($\mu, \sigma, ...$), making the figure unclear and not self-contained.

---

### Official Review · Reviewer_J9jx · 2025-10-31

**Soundness:** 2
**Presentation:** 2
**Contribution:** 2
**Rating:** 2
**Confidence:** 3

**Summary:**

The authors propose a new approach to perform test-time adaptation using neural networks which is motivated by applications in surrogate modeling.
The method involves two main subcomponents: first, a new method for subspace-respecting representation alignment, and second, a method for choosing which source statistics to reference during alignment.
The authors demonstrate their method by comparing it to a recently proposed approach on two benchmark suites of surrogate modeling problems.

**Strengths:**

Neural surrogates are a really important application with unique challenges, and studying domain adaptation in this context is important.

I like the application problems chosen, and the fact that two different surrogate modeling tasks/datasets were considered.

**Weaknesses:**

A main weakness of this article is the lack of clarity of the contributions relative to previous work.
First, Section 4.1 should be clearer about exactly what departures were made from Significant Subspace Alignment (SSA; Adachi et al 2024).
Second, a more complete ablation study demonstrating how much improvement is being made by which of the two main suggestions (from sections 4.1 and 4.2) across all the datasets would help readers understand which aspects of this work are worth incorporating in their own methods.
But more generally, the article as presently written fails to convince the reader that significant intellectual contributions have been made beyond the current state of the literature beyond the move to the surrogate modeling application domain.

Next, I did not find the exposition of the method to be clear. At a high level, Section 4.2 sounds like it's proposing something interesting by referencing D-optimality from the design literature to select which statistics to respect. Some of the high level motivation made sense to me, but I failed to follow the exact connection to how this is going to give us good statistics for alignment. At the very least, the authors should have used some of the extra 1.5 pages they still had to give additional explanation/intuition. Ideally, they would present some rigorous mathematical results motivating this approach, building on the several decades of mathematical facts built about D-optimality.

Finally, while I like the numerical applications, the experiments overall are still somewhat limited. It's good that SSA is included, but more than one competing domain adaption method would have made the experiments more convincing.

**Questions:**

1) I don't understand equation 1. $|\mathbf{W}^\top\mathbf{V}|$ is meant to be an elementwise absolute value, is that right? I initially thought it was a determinant but the matrix is not square. If it is an elementwise absolute value, what's the motivation for the squaring in the quantity $(1+|\mathbf{W}^\top\mathbf{V}|)^2$?

---

### Official Review · Reviewer_mFys · 2025-10-31

**Soundness:** 3
**Presentation:** 3
**Contribution:** 3
**Rating:** 8
**Confidence:** 3

**Summary:**

The paper introduces SATTS (Stable Adaptation at Test-Time for Simulation), a method that adapts pre-trained neural surrogates to distribution shifts during deployment without requiring costly retraining or ground-truth labels.

The core contributions are:
(i) First TTA study for neural surrogates: Introduces Test-Time Adaptation specifically for simulation and generative design optimization.
(ii) Two-part method: ATTS: A feature alignment component with automated weighting and selection across the full feature space; SATTS: An extension with online self-calibration and model selection using a quasi D-optimal criterion to select representative source statistics. Evaluation on industrial benchmarks demonstrate stable performance on SIMSHIFT and EngiBench where other TTA methods degrade

**Strengths:**

This is the first systematic exploration of TTA for high-dimensional regression in neural simulation surrogates, opening a promising research direction at the intersection of machine learning and scientific simulation.

The practical limitations of existing approaches (UDA, full retraining) are clearly articulated, addressing real industrial constraints where test configurations vary widely and source data access is restricted.

The automated feature weighting mechanism and source-free model selection strategy are clever extensions to standard feature alignment. Using a quasi D-optimal criterion for selecting representative source statistics elegantly solves the model selection problem in TTA contexts.

The evaluation is thorough, using diverse industrial benchmarks with appropriate baselines (source-only, UDA, SSA). The ablation study effectively demonstrates ATTS's advantage over methods requiring manual feature dimension selection.

**Weaknesses:**

While stability is the key achievement, quantitative improvements (e.g., RMSE) over the non-adapted baseline are often marginal in several experiments (Table 1), potentially raising questions about practical benefit versus added complexity.

The line search for model selection introduces test-time overhead. A detailed analysis of the cost-performance trade-off would strengthen the paper.

The comparison primarily focuses on SSA. Including additional recent TTA methods would provide broader context, even if originally designed for classification.

**Questions:**

- Could you provide detailed analysis of SATTS's computational overhead at inference time? How does the line search and adaptation time scale with model size or test batch count compared to performance gains?

- In EngiBench experiments, source/target domains were partitioned using t-SNE visualizations. Could this manual partitioning introduce bias? Have you considered automated split definitions to ensure generalizability?

- Could you elaborate on the gap between SATTS and UDA? What intrinsic advantages does UDA have beyond access to target inputs during pretraining, and is there a clear path for future TTA methods to close this gap?

---

### Note · Authors · 2025-11-24

**Comment:**

We thank all reviewers for their thoughtful feedback. While the reviewers found the problem relevant and the approach practically motivated, they also raised valid concerns about novelty, clarity, and performance. Given the extent of revisions needed, we have decided to withdraw the paper and will address these issues in a future submission.

The Authors

**Withdrawal Confirmation:**

I have read and agree with the venue's withdrawal policy on behalf of myself and my co-authors.